**Data Availability Statement:** There are several legal restrictions to data sharing in Norway, and this is not approved by the Norwegian Data Protection Authorities The Norwegian Social

# The association between geriatric treatment and 30-day readmission risk among medical inpatients aged ≥75 years with multimorbidity

**Marte Sofie Wang-Hansen** [1,2]*, **Hege Kersten**[3,4,5], **Jūratė Šaltytė Benth**[2,6], **Torgeir Bruun Wyller**[2,7]

1 Department of Geriatric Medicine, Vestfold Hospital Trust, Tønsberg, Norway, 2 Institute of Clinical Medicine, University of Oslo, Oslo, Norway, 3 Aging and Health, Norwegian Centre for Research, Education and Service Development, Vestfold Hospital Trust, Tønsberg, Norway, 4 Department of Research and Development, Telemark Hospital Trust, Skien, Norway, 5 Department of Pharmaceutical Bioscience, School of Pharmacy, University of Oslo, Oslo, Norway, 6 Health Services Research Unit, Akershus University Hospital, Lørenskog, Norway, 7 Department of Geriatric Medicine, Oslo University Hospital, Oslo, Norway

* marte.wang-hansen@siv.no

## Abstract

### Background

Readmission to hospital is frequent among older patients and reported as a post-discharge adverse outcome. The effect of treatment in a geriatric ward for acutely admitted older patients on mortality and function is well established, but less is known about the possible influence of such treatment on the risk of readmission, particularly in the oldest and most vulnerable patients. Our aim was to assess the risk for early readmission for multimorbid patients > 75 years treated in a geriatric ward compared to medical wards and to identify risk factors for 30-day readmissions.

### Methods

Prospective cohort study of patients acutely admitted to a medical department at a Norwegian regional hospital. Eligible patients were community-dwelling, multimorbid, receiving home care services, and aged 75+. Patients were consecutively included in the period from 1 April to 31 October 2012. Clinical data were retrieved from the referral letter and medical records.

### Results

We included 227 patients with a mean (SD) age of 86.0 (5.7) years, 134 (59%) were female and 59 (26%) were readmitted within 30 days after discharge. We found no statistically significant difference in readmission rate between patients treated in a geriatric ward versus other medical wards. In adjusted Cox proportional hazards regression analyses, lower age (hazard ratio (95% confidence interval) 0.95 (0.91–0.99) per year), female gender (2.17

Science Data service has not approved data delivery outside of Europe; consent for publication of raw data was not obtained from participants included in the study; and, importantly, complete anonymization is inachievable as the data contains potentially identifying patient information that may be trackable. For these reasons, we ask that the data are available only upon request. Data requests can be addressed to Department of Research at Vestfold Hospital Trust by Tomm Bernklev tomm.bernklev@siv.no.

**Funding:** The author(s) received no specific funding for this work.

**Competing interests:** The authors have declared that no competing interests exist.

(1.15–4.00)) and higher MMSE score (1.03 (1.00–1.06) per point) were significant risk factors for readmission.

## Conclusions

Lower age, female gender and higher cognitive function were the main risk factors for 30-day readmission to hospital among old patients with multimorbidity. We found no impact of geriatric care on the readmission rate.

## Introduction

Readmission to hospital is frequent among older patients and reported as a post-discharge adverse outcome [1]. Higher readmission rates result in increased pressure on hospital beds and higher costs, and represents a strain upon the patients. Reported readmission rates vary considerably (5–35%), depending on time span, age group and patient population [2]. Research on risk factors for readmission among patients above 75 years and those with impaired activities of daily living (ADL) is scarce [3]. In two systematic reviews [1, 4], only one study including patients older than 75 years was identified [5]. In that study, risk factors for readmission were severe disability for self-feeding and the presence of frailty markers [5].

For elderly patients the effect of hospital-based geriatric care on mortality, function and nursing home placement is well established [6–8]. Considerably less is known about the possible influence of treatment in a geriatric ward on the risk of early readmission, in particular for the oldest patients with multiple diseases and poor function. Previous reviews have not distinguished between early and late readmissions [6, 7], the share of studies on early readmissions is low [9, 10] and few patients with cognitive impairment have been included [11].

In the most recent Cochrane review of geriatric treatment with models based on comprehensive geriatric assessment (CGA) [7], only two studies of early readmission were identified [9, 10]. Furthermore, the systematic reviews on readmission risks did not include treatment in a geriatric ward as a potential explanatory variable [1, 4]. Since the evidence from randomized trials is so scarce, there is a need for observational studies addressing this relationship.

Accordingly, the aim of this study was to identify independent risk factors for readmissions within 30 days for multimorbid medical in-patients aged 75 years or more and receiving home care services, and in particular to study whether treatment in an acute geriatric ward was associated with a reduced readmission rate compared with treatment in regular medical wards.

## Materials and methods

This is a prospective cohort study with consecutive inclusion of patients acutely admitted to the medical department of Vestfold Hospital Trust and living in six of twelve municipalities in the Norwegian county Vestfold (with 151,300 inhabitants in the six target municipalities). The participating municipalities were similar to the others in terms of size, distance to hospital and population composition.

### Patient selection

During the period from April to October 2012, a project nurse screened all emergency admissions from the target municipalities to consecutively recruit patients who met the following inclusion criteria: home-dwelling, aged 75 years or more, suffering from at least two chronic

conditions and receiving municipal home care services. Patients not able to give informed consent, who were terminally ill, or who lived in a long-term care facility prior to admission were not included.

The medical department of Vestfold Hospital Trust consists of separate wards for medical subspecialties including neurology and geriatrics. The main difference between the geriatric ward and the other wards is the presence of a multidisciplinary team consisting of a physiotherapist, an occupational therapist and a pharmacist, in addition to healthcare workers, nurses and doctors, all working within the framework of a structured multiprofessional model based on the principles of CGA [12]. The characteristics of the study cohort have previously been published [13].

### Data collection

Eligible patients were approached by the project nurse as soon as possible after admission, normally the next day, and the project nurse used the Norwegian Version of the Barthel Index to assess independence in personal activities of daily living [14] and the Mini-Mental State Evaluation–Norwegian Revised Version (MMSE-NR) for cognitive screening [15]. Height and weight were measured and Body Mass Index (BMI) was calculated. A physiotherapist did an assessment of handgrip strength (HGS) within the first three days of the hospital stay. The test was completed with both hands using a Jamar dynamometer, the mean value of three trials was used. The result from the strongest hand was used for the analyses.

Two physicians, one consultant specialized in geriatric medicine and old age psychiatry and one resident in geriatric medicine, scrutinized all medical records of the included patients. Complete information regarding the current acute hospital admission, prior admissions and medications at the time of admittance was retrieved from the referral letter, electronic patient records and the patient administrative system. The Cumulative Illness Rating Scale for Geriatrics (CIRS-G) was scored for each patient [16]. All readmissions occurring within 30 days of discharge were retrieved from the patient administrative system. All readmissions to any hospital department, including psychiatric and palliative wards were counted. Deaths in the observation period was retrieved from the Norwegian Death Registry.

### Laboratory measurements

Blood samples were drawn as part of the clinical routine, and blood tests chosen based on previous reports [17, 18]. Haemoglobin (Hgb) was measured on either Sysmex XE 2100 or Sysmex XE 5000 (Sysmex Europe) with reagents from the supplier. Serum creatinine was analysed on Vitros 5.1 (Ortho-Clinical Diagnostics, USA) with reagents from the supplier. Glomerular filtration rate (GFR) was estimated using the Modification of Diet in Renal Disease equation [19].

### Statistical analyses

Demographic and clinical characteristics are presented as means and standard deviations (SD) or frequencies and percentages. Groups of patients were compared by independent samples t-test or $\chi^2$-test. Unadjusted and adjusted proportional hazards Cox regression models were estimated to assess the association between pre-selected, relevant patient characteristics and the rate of readmission. Relevant variables were selected based on clinical judgement, previous reports, correlations between the variables in the dataset, and number of missing cases for some variables. Death was considered as a competing risk factor for readmission and was included in the model as such. Schoenfeld residuals for continuous characteristics and survival curves for categorical characteristics were used to assess the assumption of proportional

hazards. Martingale residuals were used to assess the functional form of continuous characteristics. The results are presented as hazard ratios (HR) and 95% confidence intervals (CI). Eight patients were excluded from regression analyses because of one or more missing values. All tests were two-sided and results with p-values below 0.05 were considered statistically significant. For illustrative purposes, we also created Kaplan-Meier plots for treatment in the geriatric ward versus other wards as well as for all statistically significant risk factors. In these plots, age and MMSE score were dichotomized at the median.

The analyses were carried out using SPSS version 26 and STATA version 16.

### Ethics

Written informed consent was obtained from all the participants in accordance with Norwegian legal regulation. The study was presented to the Regional Committee for Medical Research Ethics and approved by the Norwegian Social Science Data Services.

## Results

A flowchart of the study sample is given in Fig 1.

The mean age of the participants was 86.0 years (SD 5.7), 134 (59%) were female and 59 (26%) were readmitted within 30 days of discharge. Geriatric ward patients were older (p = 0.009), stayed shorter at hospital (p = 0.017), had delirium more often (p = 0.022), had lower Barthel score (p = 0.031), lower BMI (p = 0.003), and higher eGFR (p = 0.026) than patients from other wards. Among the patients in geriatric ward, 3 of 52 (5.8%) died within 30 days of discharge, versus 17 of 175 (9.7%) of the patients in the other wards (p = 0.378). Patient characteristics are presented in Table 1. The readmission rate for patients treated in the geriatric ward was 16.9%, and for those treated in medical wards 25.0% (p = 0.206). A higher proportion (67%) of patients admitted to medical wards was discharged back home (as opposed to a community based rehabilitation unit) than of patients admitted to the geriatric ward (50%) (p = 0.027). Among patients discharged to a community based rehabilitation unit, 21% were readmitted, compared to 29% among those discharged home, (p = 0.130).

In unadjusted Cox models, presented in Table 2, lower age, higher MMSE score and increased number of daily drugs were significantly associated with a higher risk of readmission, while lower age, female gender and higher MMSE score were independent and significant risk factors for higher readmission rate in the adjusted model. According to analysis of Schoenfeld residuals and survival plots, only MMSE violated the proportional hazards assumption. This variable was therefore included as a time-dependent characteristic in the Cox model. No non-linearity issues were identified, as judged by martingale residuals.

Kaplan-Meier plots for the statistically significant risk factors as well as for treatment in geriatric versus other wards are displayed in Fig 2.

## Discussion

### Treatment in a geriatric ward versus other wards

The main purpose of our study was to assess whether treatment in a geriatric ward as compared with other medical wards influenced on the readmission rate in the oldest-old patients with multimorbidity. No association between the hospital ward and readmission rate was identified.

One might expect that better discharge planning and a stronger emphasis on individual treatment plans in a geriatric ward might reduce the risk of early readmission [20]. Moreover,

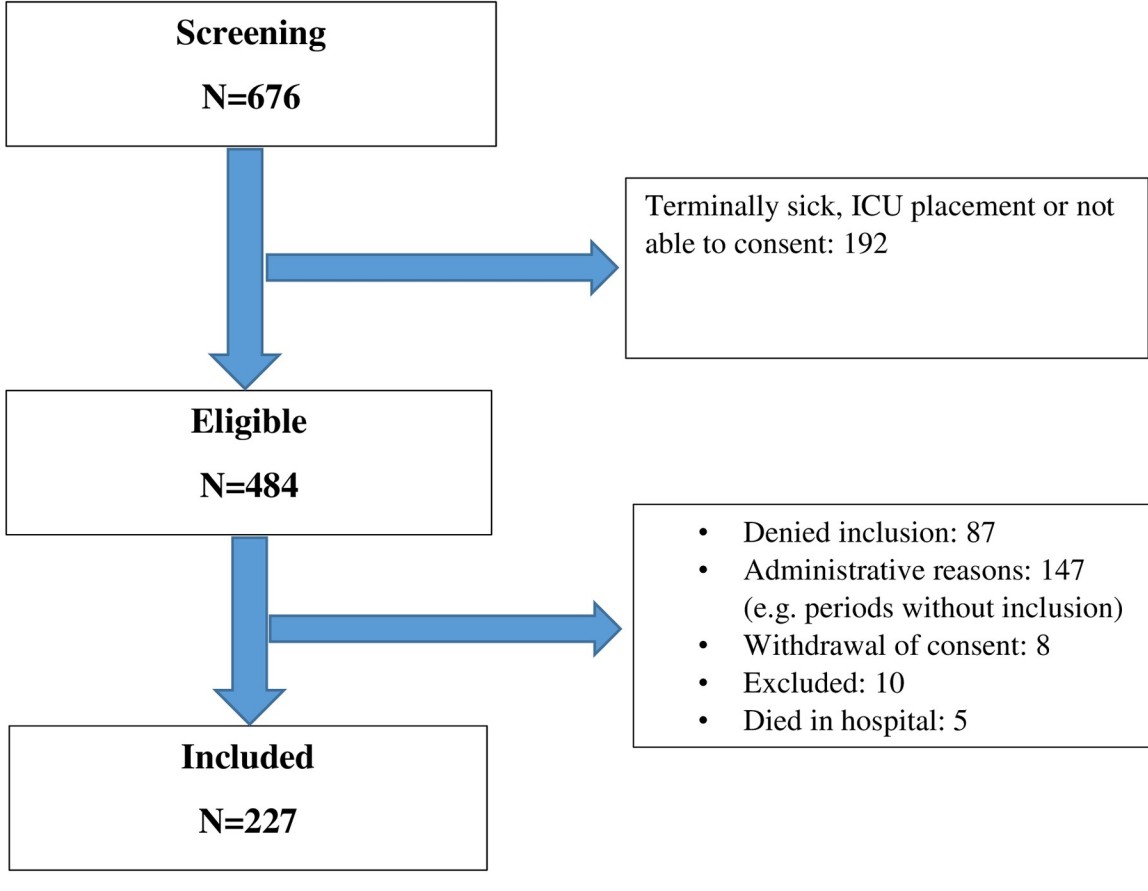

**Fig 1. Patient flow.**

routines for medication reviews, with a clinical pharmacist as part of the multidisciplinary team in the geriatric ward, could be expected to prevent early readmissions. We have previously shown that patients treated in the geriatric ward had more drug changes and were discharged with fewer potentially inappropriate drugs [21]. However, the evidence so far does not support the supposition that treatment in a geriatric ward substantially decreases early readmission rate [9–11].

As expected, patients in the geriatric ward were older, had more often delirium and were more malnourished and more functionally impaired than those admitted to the other medical wards. Such known prognostic variables were accounted for in the multivariate analysis. Due to the observational design of the study, we can however not rule out the possibility of residual confounding. Patients more likely to be readmitted for some reason not measured might have been more likely to be placed in the acute geriatric ward during the index stay. For example, patients in the geriatric ward might have fewer organ-specific symptoms and more general symptoms related to a non-specific functional decline, both conditions that have been associated with an increased readmission risk [16, 22, 23]. A larger proportion of the patients in the geriatric ward was discharged to a community rehabilitation unit. Only a few earlier studies have investigated discharge destination and risk for readmission, with one study showing higher risk for readmission for patients discharged to a nursing home [24] while a few other studies had no significant results [1].

**Table 1. Characteristics of the participants, N = 227.**

| | Patients readmitted n = 59 | Patients not readmitted n = 168 | Patients admitted to geriatric ward n = 52 | Patients admitted to medical wards n = 175 |
|---|---|---|---|---|
| Age, years. Mean (SD) | 84.0 (5.2) | 86.7 (5.8) | 87.8 (5.4) | 85.5 (5.7) |
| Female gender. N (%) | 42 (71) | 92 (55) | 31 (60) | 103 (59) |
| Geriatric ward. N (%) | 10 (16.9) | 42 (25.0) | - - - | - - - |
| Living alone. N (%) | 40 (68) | 122 (73) | 39 (75) | 123 (70) |
| Length of stay. Median (IQR) | 6.0 (3.0–9.0) | 5.0 (3.0–8.0) | 5.3 (4.0–7.8) | 6.0 (3.0–8.0) |
| Delirium during stay. N (%) | 17 (29) | 54 (32) | 23 (44) | 48 (27) |
| Barthel ADL Index. Mean (SD) n = 223 | 13.6 (4.9) | 12.9 (4.8) | 11.8 (4.1) | 13.5 (5.0) |
| BMI, kg/m$^2$. Median (IQR) n = 217 | 23.8 (20.9–27.4) | 22.7 (20.4–26.6) | 22.0 (19.0–25.2) | 23.4 (20.9–27.3) |
| MMSE. Mean (SD) n = 223 | 23.8 (5.1) | 22.8 (5.2) | 21.9 (5.7) | 23.3 (5.0) |
| HGS, kg. Mean (SD). Women n = 106 | 11.6 (5.7)[a] | 10.6 (4.5)[a] | 9.7 (5.6)[c] | 11.3 (4.7)[c] |
| HGS, kg. Mean (SD). Men n = 75 | 24.8 (9.7)[b] | 20.8 (8.4)[b] | 20.5 (8.3)[d] | 21.9 (8.9)[d] |
| Number of daily medications. Mean (SD) | 8.7 (3.6) | 7.7 (3.6) | 7.4 (2.7) | 8.1 (3.9) |
| CIRS-G total score. Mean (SD) | 22.2 (5.2) | 21.3 (6.3) | 22.0 (6.4) | 21.4 (5.9) |
| Haemoglobin, g/100 mL. Mean (SD) | 10.4 (1.9) | 11.4 (1.8) | 11.2 (1.6) | 11.0 (1.9) |
| eGFR. Mean (SD) n = 226 | 45.6 (15.3) | 48.2 (14.7) | 51.2 (12.8) | 46.4 (15.3) |
| Alive after 12 months. N (%) | 27 (46) | 111 (66) | 34 (65) | 104 (59) |

ADL = Activities of Daily Living, BMI = Body Mass Index, MMSE = Mini Mental State Evaluation—Norwegian Revised Version, HGS = Hand Grip Strength,

CIRS-G = Cumulative Illness Rating scale for Geriatrics, eGFR = Estimated Glomerular Filtration Rate.

*Chi-square for categorical variables. Independent sample t-test for continuous variables.

[a]Females: 35 patients readmitted and 71 not readmitted.

[b]Males: 14 patients readmitted and 61 not readmitted.

[c]Females: 22 patients geriatric ward and 84 patients medical ward.

[d]Males: 19 patients geriatric ward and 56 patients medical ward.

**Table 2. Results of Cox proportional hazards regression analysis of risk of readmission with death as «competing risk».** N = 219.

| Variable | Unadjusted | | Adjusted | |
|---|---|---|---|---|
| | HR (95% CI) | p-value | HR (95%) | p-value |
| Age | 0.94 (0.90; 0.98) | **0.005** | 0.95 (0.91; 0.99) | **0.019** |
| Gender, female | 1.75 (0.98; 3.13) | 0.058 | 2.17 (1.15; 4.00) | **0.016** |
| Barthel ADL | 1.03 (0.97; 1.09) | 0.331 | 1.02 (0.96; 1.08) | 0.616 |
| CIRS-G | 1.03 (0.99; 1.07) | 0.159 | 1.03 (0.97; 1.10) | 0.309 |
| eGFR | 0.99 (0.98; 1.01) | 0.434 | 1.00 (0.98; 1.02) | 0.909 |
| Ward (Geriatrics) | 0.64 (0.31; 1.34) | 0.240 | 0.78 (0.36; 1.73) | 0.545 |
| MMSE (time dependent) | 1.03 (1.01; 1.06) | **0.013** | 1.03 (1.00; 1.06) | **0.034** |
| Number of daily drugs | 1.08 (1.01; 1.16) | **0.035** | 1.04 (0.95; 1.13) | 0.421 |

ADL = Activities of Daily Living, MMSE = Mini Mental State Examination—Norwegian Revised Version,

CIRS-G = Cumulative Illness Rating scale for Geriatrics, eGFR = Estimated Glomerular Filtration Rate.

Significant Differences significant at 0.05 level in bold font.

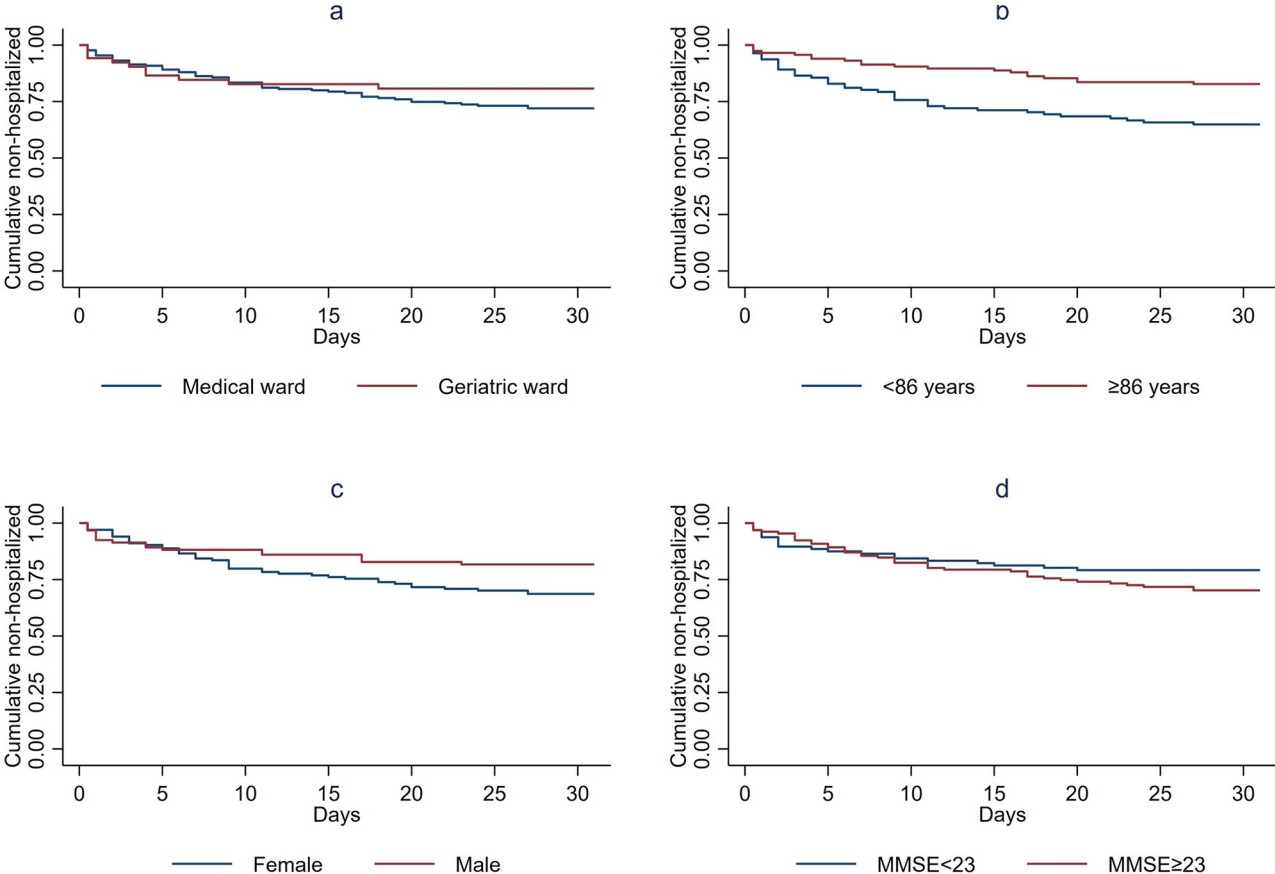

**Fig 2.** Kaplan-Meier plots for one-year survival by a) Ward, b) Age, b) Gender and d) MMSE score. a) Ward, n = 227, no cases were censored. b) Age, n = 227, no cases were censored. c) Gender, n = 227, no cases were censored. d) MMSE score, n = 223, no cases were censored.

## Risk factors for readmission

Lower age, female gender and higher MMSE score were independently associated with higher readmission risk. These results may be considered as surprising, since studies of younger populations among other factors have found male gender, higher age, poor overall health and functional disability to be associated with higher risk for early readmission [1, 25]. We are concerned that these results may reflect elements of ageism in the Norwegian health care system that prevent the oldest patients and those with cognitive impairment from being readmitted to hospital even when they might benefit from it. In Norway, strong financial incentives have been introduced, aiming at transferring emergency treatment of frail elderly patients to a lower level of care. Our results may indicate that older patients with signs of cognitive decline now tend to be treated acutely in nursing home beds [26] instead of being readmitted to hospital when exacerbations of their chronic disorders occur. Whereas treatment in geriatric wards is firmly evidence based, this is not the case for emergency nursing home beds. A reluctance to readmit the oldest and most severely cognitively impaired patients to hospital may thus hinder them of receiving the most effective treatment.

In younger patient groups, previous research has found higher readmission rates among men than among women [1, 25]. For older populations, previous research, like ours, identified female gender as a risk for readmission [27].

Taken together, our results indicate that vulnerable elderly patients may have different risk factors for early readmission than younger and more robust populations.

## Readmission rate

The early readmission rate was high in our material (26%) compared to several previous studies [2, 25]. Most of the available literature on early readmission is based on studies of younger patients who have other characteristics and social needs.

We identified only one study reporting higher rates of early readmission than we found, and that was a study including patients at high risk of hospitalization [2]. Our high readmission rate might be explained by the characteristics of our study cohort such as a high mean age, pronounced comorbidity and inclusion of patients with mild to moderate cognitive impairment. Moreover, all patients received home care service, indicating some degree of ADL impairment. As previously reported [28], our patients had a high prevalence of sarcopenia, which is an important frailty marker. All these factors are known to increase the risk of readmission [29–32].

## Strengths and limitations

Only very few studies of readmissions have comprised patients aged 75 years or more [1]. Our study thus contributes to a field where there is little research-based knowledge.

Norwegian hospitals are financed by the government. Patients in the index municipalities use Vestfold Hospital Trust regardless of disease, and follow-up of the patients regarding readmission was therefore complete. We also had access to complete data on deaths through the Norwegian Death Registry. We were therefore able to include death within 30 days after discharge as a competing risk.

An important weakness is the observational design, preventing us from drawing any firm causal inference. Unknown and unmeasured confounding factors may have been present. The study had insufficient power to include more covariates in the regression analysis, which might have been desirable. On the other hand, preliminary correlation analyses (not shown) indicated that inclusion of other variables into the adjusted model would not have provided more information. The fact that more patients from the geriatric ward were discharged to municipal rehabilitation units, and patients discharged to such units to a lesser degree were readmitted, might be considered a confounding factor. We will, however, argue that better identification of rehabilitation needs and more adequate use of post discharge rehabilitation units constitute integral parts of geriatric care, and as so is not reasonable to adjust for.

The power may have been insufficient for detecting differences in readmission rate between the wards. If the real difference is as low as 16.9% versus 25.0%, as our estimates suggest, a considerably larger sample would presumably be needed to get a statistically significant result. Moreover, we do not know whether reasons for exclusion differed between the groups (Fig 1). Different selection mechanisms may have been active in the geriatric than in then general medical group, thus potentially influencing the external validity of our findings. But since the potentially confounding variables have been measured, we have been able to adjust for them in the multivariate analyses. Accordingly, we presume that the results of the adjusted analyses have god validity. Patients with severe dementia, were excluded due to the requirement for informed consent. In Vestfold Hospital Trust these patients are treated in all medical wards, so the exclusion would not influence the results for different wards.

Another possible limitation is that we registered all-cause readmissions to the same hospital without discriminating between acute and elective cases. Other studies have had different approaches regarding this issue, with some including only readmissions for the same medical

problem while others included all-cause readmissions [1]. In a study carried out to evaluate a community-based discharge scheme, most of the readmissions were emergency readmissions, and they were especially prominent among patients older than 85 years [27].

## Conclusion

Lower age, female gender and higher cognitive function were the main risk factors for 30-day readmission to hospital, but no differences between geriatric and other medical wards were identified.

## Acknowledgments

We would like to thank Morten Lindberg, senior consultant at the Vestfold Hospital Trust central laboratory, and the laboratory for carrying out the laboratory tests and providing information about the analyses used; Lara T. Hvidsten for help with performing CIRS-G; and Gløer Gløersen for collecting all information regarding medications.

## Author Contributions

**Conceptualization:** Marte Sofie Wang-Hansen, Hege Kersten, Torgeir Bruun Wyller.

**Data curation:** Marte Sofie Wang-Hansen.

**Formal analysis:** Marte Sofie Wang-Hansen, Jūratė Šaltytė Benth.

**Methodology:** Torgeir Bruun Wyller.

**Writing – original draft:** Marte Sofie Wang-Hansen, Hege Kersten, Jūratė Šaltytė Benth, Torgeir Bruun Wyller.

**Writing – review & editing:** Marte Sofie Wang-Hansen, Hege Kersten, Jūratė Šaltytė Benth, Torgeir Bruun Wyller.

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
