## [Decision Letter · Decision Letter 0]

3 Aug 2021

PONE-D-21-13191

The association between geriatric treatment and 30-day readmission risk among elderly medical inpatients

PLOS ONE

Dear Dr. Wang-Hansen,

Thank you for submitting your manuscript to PLOS ONE. After careful consideration, we feel that it has merit but does not fully meet PLOS ONE’s publication criteria as it currently stands. Therefore, we invite you to submit a revised version of the manuscript that addresses the points raised during the review process.

ACADEMIC EDITOR: 

Please consider the methodological aspects raised by reviewers, including  the issue of “municipal emergency beds”, that actually sound as a different form of hospital admission, and comment on cases selection process (Figure 1), leading to the inclusion of a minority of patient: are these data generalizable?

A further aspects that I feel is lacking are is a statistical comparison of patients’ features between Geriatrics and Internal Medicine group, as the former looks, as expected, older and more cognitively impaired. Can this difference bias results? Moreover a discussion on statistical power should be included among study limitations, as the numerical difference observed in readmissions between the two groups might be non significant due to small sample size.

We look forward to receiving your revised manuscript.

Kind regards,

Enrico Mossello

Academic Editor

PLOS ONE

Additional Editor Comments (if provided):

Reviewers' comments:

Reviewer's Responses to Questions

**Comments to the Author**

1. Is the manuscript technically sound, and do the data support the conclusions?

Reviewer #1: No

Reviewer #2: Partly

2. Has the statistical analysis been performed appropriately and rigorously? 

Reviewer #1: I Don't Know

Reviewer #2: Yes

3. Have the authors made all data underlying the findings in their manuscript fully available?

Reviewer #1: Yes

Reviewer #2: No

4. Is the manuscript presented in an intelligible fashion and written in standard English?

Reviewer #1: No

Reviewer #2: Yes

5. Review Comments to the Author

Reviewer #1: introduction

line 58: It is unclear what unsatisfactory patient pathways means. this could use some clarification

line 60: what is meant by the oldest patients? What age group specifically?

line 64: There is somewhat of a disconnect between the ideas in paragraph 1 and 2. I think there is an opportunity to better link the need for looking at readmission risk factors in older individuals and CGA as a factor.

line 68: but there have been some studies that have looked at CGA and readmission. What are you contributing that they did not do?

lines 72-73: The purpose does not seem to link back to any of the old-old age group discussed in the first paragraph. need better/clearer connection between what is known, gaps, and purpose.

methods:

line 78: I am confused with how the participants were recruited. While they were admitted or from some other community database that had information on individuals who were recently admitted to the hospital. can you please clarify this in the manuscript. This also seems to be participants from a larger study. please describe more about the larger study design and who you are selecting from that larger study.

line 93: I think that participants need to be receiving home care services is lost earlier in this manuscript. That should be included when describing your population and perhaps considered in the introduction and discussion sections. what type of home care are these individuals receiving should also be clarified.

data collection and lab measures: It is a little unclear here why you are collecting the data you are collecting at specific time points. perhaps a conceptual model or other table or figure would clarify why you are collecting specific data and certain times.

results: I feel like the CGA piece is completing lost in this section. How do these findings fit with CGA versus just looking at readmission risk factors. I am very confused about the focus of this manuscript

line 146: I am confused. they needed to be receiving home care to be included, but some were also receiving care in a nursing home? Do these contradict each other?

discussion: This needs to be re-focused around what is the main purpose and variables of interest for your study. There are multiple focuses currently and it is confusing to read. is it focused on readmission, readmission risk, old-old individuals, CGA? There are a lot of ideas and many of these have been already covered in previous research. You will also want to work on what is new here? what is the gap you are addressing that we haven’t already examined?

Reviewer #2: Wang-Hansen et al. report the risk of readmission from a single Norwegian hospital following discharge, for older adults aged over 75 years old and receiving home care services. The consented study population is divided according to management in either a general medical or specialist geriatric medicine setting, the latter of which is used as a surrogate for receipt of comprehensive geriatric assessment. In adjusted Cox proportional hazards modelling accounting for the competing risk of death after discharge, geriatric medicine care was not associated with any difference in the risk of readmission. The authors found that younger, female patients with higher cognitive function were at the highest risk of 30-day readmission.

The manuscript addresses a topic of importance which is not well covered in the wider literature. However, there are some areas of the analysis that are concerning and I have raised these points below.

1. A major concern is about population generalisability to “geriatric medicine care” and by association “comprehensive geriatric assessment”. Some elements of this are limitations of a consented observational cohort, but these should still be acknowledged in the discussion. The requirement for informed consent prevents many patients with delirium and dementia from participation; this group would usually make up a large part of geriatric medicine care. It would be helpful within Figure 1 to breakdown the numbers for the loss from ineligibility by patients managed in general medicine and geriatric medicine settings – it is important for the reader to know if the group representing geriatric medicine in this analysis only account for a minority of patients in the wards.

2. Similarly for transparency, it would be helpful to know how many deaths occurred after discharge within 30 days and in which groups (general medicine or geriatric medicine) these occurred. Currently the table only provides survival to 1 year. The use of a competing risk Cox model is welcome, but given the small sample size, applicability may be questionable if many more deaths occurred within the modelling window for the small geriatric medicine group of just 52 patients.

3. I also have concerns about the patient group here as the discharge destination numbers seem very surprising. The inclusion criteria require all patients to have been admitted from home. In the geriatric medicine managed group, 50% are then discharged to a nursing home following a median length of stay of just 5 days. This seems very unlikely if such a transfer was a permanent switch of residence, particularly since 33% of general medical patients followed the same route. I am concerned that these are “discharges” to rehabilitation facilities for post-acute care with a later assessment of return home. This would clearly lower the risk of hospital readmission, as such facilities often have access to nursing and/or medical care that would not otherwise be available if the patient had been discharged to their usual residence. Can the authors explain if this is the case, and if so acknowledge this as a bias that might explain the direction of their results?

4. As well as this, the discussion raises a concern about the outcome ascertainment: “Our results may indicate that older patients with signs of cognitive decline now tend to be treated in municipal emergency beds…”. I am unfamiliar with the health service setting here, but transfer from a home to a “municipal emergency bed” sounds like it is not included in the outcome measure, but is a (re)admission to a healthcare setting? If so, this is a significant limitation to interpretation of the results. Can I suggest the authors provide some context for readers not familiar with the Norwegian system and explain this a bit clearer in the discussion?

5. In Figure 1, the majority of eligible patients excluded from the study were due to “administrative reasons” rather than lack of consent. This is not explained in the methods but is particularly important because the term “consecutive patients” is used to imply no selection bias in approach of eligible patients. Can the authors clarify what administrative reasons means?

6. PLOS authors have the option to publish the peer review history of their article (what does this mean?). If published, this will include your full peer review and any attached files.

Reviewer #1: No

Reviewer #2: No

---

## [Author Response · Author response to Decision Letter 0]

21 Oct 2021

Point-by-point response to reviewer comments for manuscript PONE-D-21-13191

“The association between geriatric treatment and 30-day readmission risk among elderly medical inpatients.”

Thank you for your very thorough and relevant comments.

Below we have responded point-by-point to the comments, also describing the subsequent revisions of the manuscript. 

Please consider the methodological aspects raised by reviewers, including the issue of “municipal emergency beds”, that actually sound as a different form of hospital admission, and comment on cases selection process (Figure 1), leading to the inclusion of a minority of patient: are these data generalizable? 

We agree that our wording regarding “municipal emergency beds” might cause confusion, and have rephrased the relevant section (page 14-15, line 223-230) accordingly. What has been called “municipal emergency beds” in Norway are nursing home beds open for acute admittance. These facilities are sparsely fitted, and do not regularly offer investigations such as x-ray and microbiology. If seniors in need for readmission are admitted to such beds instead of hospital beds, it may explain our findings of a decreased readmission rate with increasing age. Given the low standard of medical service connected to the “municipal emergency beds” compared to regular hospital beds, we feel that it may also be pertinent to speculate whether this is indicative of ageism. 

The patients recruited for this study were all regular medical patients and prospectively included after admission to hospital. Patients from other municipalities than those selected a priori, were not eligible and should not be included in Figure 1. We have redesigned figure 1 to make this clearer. 

A further aspects that I feel is lacking are is a statistical comparison of patients’ features between Geriatrics and Internal Medicine group, as the former looks, as expected, older and more cognitively impaired. Can this difference bias results? Moreover a discussion on statistical power should be included among study limitations, as the numerical difference observed in readmissions between the two groups might be non significant due to small sample size. 

Yes, it is correct that since the patients were not randomly allocated to the alternative wards, systematic differences between the groups do exist. We have now described this more explicitly in the Results section page 8, line 149-152. Since these potentially confounding variables have been measured, we have been able to adjust for them in the multivariate analyses. Accordingly, we presume that the results of the adjusted analyses have good validity. 

Regarding the limited statistical power, we have now added a brief discussion on this important limitation (page 16, line 266-269)

In your Data Availability statement, you have not specified where the minimal data set underlying the results described in your manuscript can be found. PLOS defines a study's minimal data set as the underlying data used to reach the conclusions drawn in the manuscript and any additional data required to replicate the reported study findings in their entirety. All PLOS journals require that the minimal data set be made fully available. For more information about our data policy, please see http://journals.plos.org/plosone/s/data-availability. 

There are several legal restrictions to data sharing in Norway, and this is not approved by the Norwegian Data Protection Authorities The Norwegian Social Science Data service has not approved data delivery outside of Europe; consent for publication of raw data was not obtained from participants included in the study; and, importantly, complete anonymization is unachievable as the data contains potentially identifying patient information that may be trackable. For these reasons, we ask that the data are available only upon request. Data requests can be addressed to Department of Research at Vestfold Hospital Trust by Tomm Bernklev tomm.bernklev@siv.no. 

Reviewer #1 

line 58: It is unclear what unsatisfactory patient pathways means. this could use some clarification. 

We have reformulated the sentence: Page 4, line 55.

line 60: what is meant by the oldest patients? What age group specifically? 

We have clarified in the titlr that our focus is patients aged 75+: Page 1, line 3 and page 3, line 26.

line 64: There is somewhat of a disconnect between the ideas in paragraph 1 and 2. I think there is an opportunity to better link the need for looking at readmission risk factors in older individuals and CGA as a factor. 

We agree, and we have clarified treatment in a geriatric ward as one of many factors that may affect readmission rate. Page 4, line 62 – 64. 

line 68: but there have been some studies that have looked at CGA and readmission. What are you contributing that they did not do? 

We have rephrased the entire text and used the term geriatric care instead of CGA, as we think that this is most accurate for what we have studied. CGA was an element of the treatment in our geriatric ward, but this is not a study of CGA per se. We have explained in the Introduction (page 5, line 68-70) that the most recent Cochrane review includes two studies of readmission including elements of geriatric care as a potentially explanatory factor (White et al 1994 (reference #9) and Wald et al 2011 (reference #10)). These studies were small, and did not investigate a geriatric ward but a CGA team without geriatrician.

lines 72-73: The purpose does not seem to link back to any of the old-old age group discussed in the first paragraph. need better/clearer connection between what is known, gaps, and purpose. 

We have rephrased the text to better clarify our emphasis on the oldest old. Page 5, line 75.

line 78: I am confused with how the participants were recruited. While they were admitted or from some other community database that had information on individuals who were recently admitted to the hospital. can you please clarify this in the manuscript. This also seems to be participants from a larger study. please describe more about the larger study design and who you are selecting from that larger study. 

We agree that our phrasing was confusing on this point, and are sorry for that! The participants were not recruited form any database or larger study, but were all prospectively included to this specific study after admission to hospital. We have redesigned figure 1 and rephrased the first part of the Materials and methods section (page 5, line 79-83) to make this clearer. 

line 93: I think that participants need to be receiving home care services is lost earlier in this manuscript. That should be included when describing your population and perhaps considered in the introduction and discussion sections. what type of home care are these individuals receiving should also be clarified. 

Thank you for this valuable advice! We have moved this information further up in the Methods section page 5, line 85-90. 

data collection and lab measures: It is a little unclear here why you are collecting the data you are collecting at specific time points. perhaps a conceptual model or other table or figure would clarify why you are collecting specific data and certain times. 

We have tried to clarify this in the text. Page 6, line 104 and page 7, line 117-118. 

I feel like the CGA piece is completing lost in this section [results]. How do these findings fit with CGA versus just looking at readmission risk factors. I am very confused about the focus of this manuscript 

We agree that the aim was unsatisfactorily described, and have changed this accordingly. For clarity, we have now consistently used “treatment in a geriatric ward” contrasted by “treatment in other wards” as relevant exposure variable throughout the manuscript. We have clarified in the Methods section (page 6, line 92-96) that treatment in the geriatric ward comprised an element of CGA, but the study is focused on the effect of ward allocation, not on the CGA methodology per se, in line with most of the RCTs of effects of geriatric treatment. 

line 146: I am confused. they needed to be receiving home care to be included, but some were also receiving care in a nursing home? Do these contradict each other? 

We have tried to clarify this se page 14, line 211-215. Patients had to stay in their own home and receive home care before the index hospital stay. After the index stay, some were discharged to nursing home based rehabilitation units. 

This [the Discussion] needs to be re-focused around what is the main purpose and variables of interest for your study. There are multiple focuses currently and it is confusing to read. is it focused on readmission, readmission risk, old-old individuals, CGA? There are a lot of ideas and many of these have been already covered in previous research. You will also want to work on what is new here? what is the gap you are addressing that we haven’t already examined? 

Large parts of the Discussion section have now been re-written, attempting to meet these pertinent comments from the reviewer. The discussion is now structured around i) the potential effect of geriatric care on readmission, ii) risk factors for readmission in general among patients aged 75+, and iii) readmission rate.

Reviewer #2: 

The manuscript addresses a topic of importance which is not well covered in the wider literature. However, there are some areas of the analysis that are concerning and I have raised these points below.

1. A major concern is about population generalisability to “geriatric medicine care” and by association “comprehensive geriatric assessment”. Some elements of this are limitations of a consented observational cohort, but these should still be acknowledged in the discussion. The requirement for informed consent prevents many patients with delirium and dementia from participation; this group would usually make up a large part of geriatric medicine care. It would be helpful within Figure 1 to breakdown the numbers for the loss from ineligibility by patients managed in general medicine and geriatric medicine settings – it is important for the reader to know if the group representing geriatric medicine in this analysis only account for a minority of patients in the wards. 

We agree that Figure 1 in its original version was confusing, as it comprised patients that were definitely ineligible. These are now removed and we have tried to make the figure clearer. However, among those who were in principle eligible, we do unfortunately not have data on the reasons for exclusion specified for each ward (geriatric versus other). This is a weakness mainly challenging external validity of the results, and we have now acknowledged this weakness in the Discussion section (page 16, line 269-271). 

2.Similarly for transparency, it would be helpful to know how many deaths occurred after discharge within 30 days and in which groups (general medicine or geriatric medicine) these occurred. Currently the table only provides survival to 1 year. The use of a competing risk Cox model is welcome, but given the small sample size, applicability may be questionable if many more deaths occurred within the modelling window for the small geriatric medicine group of just 52 patients. 

Data for 30-days mortality by ward type is given below:

Ward N (patients) %

Geriatric 3 (52) 5.8

Medical 17 (175) 9.7

The 30-days mortality was lower among patients treated in the Geriatric ward, but the difference was not statistically significant (p=0.378). We have added this information page 8, line 152-153.

3. I also have concerns about the patient group here as the discharge destination numbers seem very surprising. The inclusion criteria require all patients to have been admitted from home. In the geriatric medicine managed group, 50% are then discharged to a nursing home following a median length of stay of just 5 days. This seems very unlikely if such a transfer was a permanent switch of residence, particularly since 33% of general medical patients followed the same route. I am concerned that these are “discharges” to rehabilitation facilities for post-acute care with a later assessment of return home. This would clearly lower the risk of hospital readmission, as such facilities often have access to nursing and/or medical care that would not otherwise be available if the patient had been discharged to their usual residence. Can the authors explain if this is the case, and if so acknowledge this as a bias that might explain the direction of their results? 

The reviewer raises a relevant point, and the suggested pathway is right. “Nursing home” as discharge destination does mainly imply a community rehabilitation unit for post-acute care. Usually the patients are admitted there for 1-2 weeks and then returned to their own homes. This discharge destination was more common in the geriatric group which is now explained on page 8, line 155-158. In the Discussion, we have elaborated on this and argued that this is not a bias but bay be an effect of better identification of rehabilitation needs in the geriatric care group.

4. As well as this, the discussion raises a concern about the outcome ascertainment: “Our results may indicate that older patients with signs of cognitive decline now tend to be treated in municipal emergency beds…”. I am unfamiliar with the health service setting here, but transfer from a home to a “municipal emergency bed” sounds like it is not included in the outcome measure, but is a (re)admission to a healthcare setting? If so, this is a significant limitation to interpretation of the results. Can I suggest the authors provide some context for readers not familiar with the Norwegian system and explain this a bit clearer in the discussion? 

We understand the need for further explanation of the Norwegian context. Please see our response to the Editor above.

5. In Figure 1, the majority of eligible patients excluded from the study were due to “administrative reasons” rather than lack of consent. This is not explained in the methods but is particularly important because the term “consecutive patients” is used to imply no selection bias in approach of eligible patients. Can the authors clarify what administrative reasons means? 

Figure 1 is changed in the revised manuscript. Please see our response to the Editor above.

---

## [Editor Report · Decision Letter 1]

22 Dec 2021

The association between geriatric treatment and 30-day readmission risk among medical inpatients aged ≥75 years with multimorbidity.

PONE-D-21-13191R1

Dear Dr. Wang-Hansen,

We’re pleased to inform you that your manuscript has been judged scientifically suitable for publication and will be formally accepted for publication once it meets all outstanding technical requirements.

Kind regards,

Enrico Mossello

Academic Editor

PLOS ONE
---

## [Editor Report · Acceptance letter]

31 Dec 2021

PONE-D-21-13191R1 

The association between geriatric treatment and 30-day readmission risk among medical inpatients aged ≥75 years with multimorbidity. 

Dear Dr. Wang-Hansen:

I'm pleased to inform you that your manuscript has been deemed suitable for publication in PLOS ONE. Congratulations! Your manuscript is now with our production department. 

Kind regards, 

on behalf of

Dr. Enrico Mossello 

Academic Editor

PLOS ONE